# Shot noise distinguishes Majorana fermions from vortices injected in the edge mode of a chiral $p$-wave superconductor

C. W. J. Beenakker and D. O. Oriekhov

Instituut-Lorentz, Universiteit Leiden, P.O. Box 9506, 2300 RA Leiden, The Netherlands

September 2020

## Abstract

The chiral edge modes of a topological superconductor support two types of excitations: fermionic quasiparticles known as Majorana fermions and $\pi$-phase domain walls known as edge vortices. Edge vortices are injected pairwise into counter-propagating edge modes by a flux bias or voltage bias applied to a Josephson junction. An unpaired edge mode carries zero electrical current on average, but there are time-dependent current fluctuations. We calculate the shot noise power produced by a sequence of edge vortices and find that it increases logarithmically with their spacing — even if the spacing is much larger than the core size so the vortices do not overlap. This nonlocality produces an anomalous $V \ln V$ increase of the shot noise in a voltage-biased geometry, which serves as a distinguishing feature in comparison with the linear-in-$V$ Majorana fermion shot noise.

# 1 Introduction

A chiral $p$-wave superconductor is the superconducting counterpart to a quantum Hall insulator [1]: Both are two-dimensional materials with a gapped bulk and gapless modes that circulate unidirectionally (chirally) along the boundary. Backscattering is suppressed when the counterpropagating edge modes are widely separated. The resulting unit transmission probability for quasiparticles injected into an edge mode implies a quantized thermal conductance for both systems — half as large in the superconductor because the quasiparticles are Majorana fermions [2–4] (coherent superpositions of electrons and holes) rather than the Dirac fermions (independent electrons and holes) of an integer quantum Hall edge mode.

This close correspondence [5] between topological insulators, as in the integer quantum Hall effect, and topological superconductors, as in chiral $p$-wave superconductivity, refers to their fermionic quasiparticle excitations. The superconducting phase allows for an additional collective degree of freedom, a winding of the phase field forming a vortex, with non-Abelian rather than fermionic exchange statistics [3, 6]. Vortices are typically immobile, pinned to defects in the bulk, but they may also be mobile phase boundaries in the edge mode. The $2\pi$ winding of the superconducting phase around a bulk vortex corresponds on the edge to a $\pi$-phase domain wall for Majorana fermions [7].

It is the purpose of this work to identify electrical signatures of edge vortices, and to distinguish these from the familiar electronic transport properties of Majorana fermions [8–16]. For that purpose we contrast the two injection geometries shown in Fig. 1. Majorana fermions are injected by a voltage source, contacted via a tunnel junction to an edge mode. The analogous edge vortex injector is a flux-biased Josephson junction. A $2\pi$ increment of the superconducting phase difference $\phi$ injects one vortex into each of the opposite edges [17].

If the edge modes would propagate in the same direction, the vortices could fuse in a

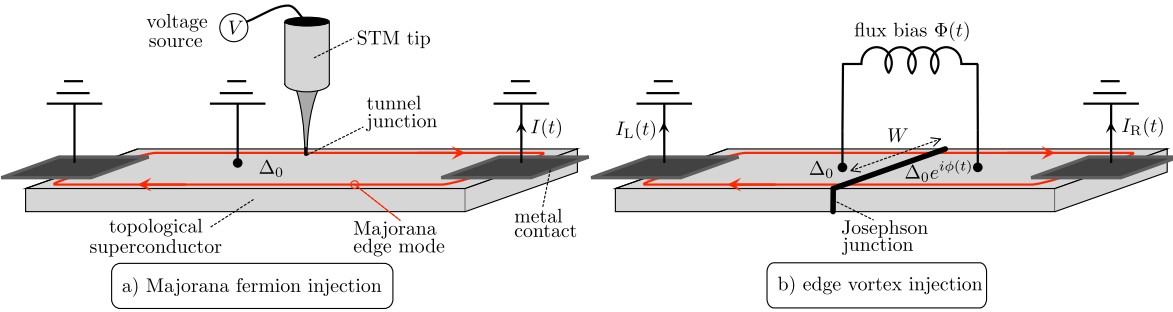

Figure 1: Topological superconductor with chiral Majorana edge modes. In panel a) a voltage bias across a tunnel junction injects Majorana fermions into the right-moving edge mode. In panel b) a flux bias across a Josephson junction injects edge vortices in the counter-propagating edge modes. The two injection processes can be detected and distinguished by shot noise measurements.

metal contact [18]. This fusion process is associated with a noiseless charge transport of $\pm e/2$ [19, 20]. (The sign depends on how the world lines of the vortices are braided.) For counterpropagating edge modes as in Fig. 1 the vortices cannot fuse, they will enter different contacts to the left and to the right of the Josephson junction. The charge transfer into each contact is zero on average, but it is not noiseless: The injection process produces shot noise, in the case of edge vortices as well in the case of Majorana fermions.

The equal-weight electron-hole superposition that is characteristic of a Majorana fermion results in a charge variance of $e^2$ per injected fermion, producing a quantized shot noise power [21]. We find that the charge variance per edge vortex is nonlocal, it depends logarithmically on the separation $L$ between pairs of vortices on the same edge:

$$\operatorname{Var} Q_{\text{vortex}} = \frac{e^2}{\pi^2} \ln(L/\lambda), \ \text{ for } \ L \gg \lambda. \tag{1.1}$$

Here $\lambda$ is the width of the $\pi$-phase domain wall, which sets the size of the edge vortex core. The dependence on the ratio $L/\lambda$ persists when $L \gg \lambda$, so when the domain walls do not overlap. This nonlocality signals the long-range correlation that exists between vortices in a topological superconductor.

The outline of this paper is as follows. In the next section we formulate the general scattering theory on which our analysis is based. The Majorana nature of the quasiparticle excitations implies that expectation values of pairs of creation operators do not vanish — as they would for Dirac fermions. This technical complication plays no role for DC transport, but needs to be accounted for in the case of time dependent perturbations, when inelastic scattering plays a role [22]. In Sec. 3 we generalize a relationship between the charge variance and the average particle current derived in Ref. 21 for DC transport to the time dependent setting. The charge noise of the edge vortices is calculated in Sec. 4 and compared with the known result [23] for Majorana fermions in Sec. 5. We propose a voltage-biased geometry in which the edge vortices produce a shot noise power that increases $\propto V \ln V$ — in contrast to the linear voltage dependence of the Majorana fermion noise power.

## 2 Trace formula for the variance of the transferred charge

We start with a general inelastic scattering formulation, in terms of a set of fermionic quasiparticle operators $a_n(E)$ for the incoming modes and $b_n(E)$ for the outgoing modes, related by the energy dependent scattering matrix,

$$b_n(E) = \int_{-\infty}^{\infty} \frac{dE'}{2\pi} \sum_m S_{nm}(E, E') a_m(E'). \tag{2.1}$$

Each mode index $n = 1, 2, \ldots N$ contains an electron and hole component in a Nambu spinor. Pauli matrices $\sigma_x, \sigma_y, \sigma_z$ act on the spinor degree of freedom (with $\sigma_0$ the $2 \times 2$ unit matrix). The scattering matrix is unitary and constrained by particle-hole symmetry,

$$S(E, E') = \sigma_x S^*(-E, -E') \sigma_x. \tag{2.2}$$

We seek the charge transferred by quasiparticle excitations at $E > 0$ into a subset $M$ of the $N$ electron-hole modes. The projector $\mathcal{D}_M$ selects these $M$ modes and the projector $\mathcal{P}_+$

selects positive energies. The charge operator for the outgoing modes is

$$Q = e \int_0^\infty \frac{dE}{2\pi} \sum_{n=1}^{M} b_n^\dagger(E) \sigma_z b_n(E) \equiv e b^\dagger \sigma_z \mathcal{D} \mathcal{P}_+ b. \tag{2.3}$$

The scattering matrix converts this into an expression in terms of the incoming mode operators,

$$Q = e a^\dagger \cdot S^\dagger \sigma_z \mathcal{D} \mathcal{P}_+ S \cdot a. \tag{2.4}$$

In these equations the Pauli matrix $\sigma_z$ accounts for the opposite charge $\pm e$ of the electron and hole components of the Nambu spinor. (For ease of notation we will set $e \equiv 1$ in many of the equations.)

Moments of $Q$ are evaluated by taking pairwise contractions of $a, a^\dagger$, each of which are given by the Fermi function $f(E)$,

$$\langle a_n^\dagger(E) a_m(E') \rangle = f(E) \sigma_0 \delta_{nm} \delta(E - E'), \quad \langle a_n^\dagger(E) a_m^\dagger(E') \rangle = f(E) \sigma_x \delta_{nm} \delta(E + E'). \tag{2.5}$$

The second contraction is anomalous [22], it does not vanish because of the particle-hole symmetry relation $a(E) = \sigma_x a^\dagger(-E)$. If the scattering is elastic the anomalous contraction which couples $+E$ to $-E$ does not contribute — but in the more general case of inelastic scattering it cannot be ignored for any moment higher than the first.

In the zero-temperature limit the Fermi function $f(E) = (1 + e^{E/k_B T})^{-1}$ becomes a projector $\mathcal{P}_-$ onto negative energies. We will take that limit in what follows. This also means that thermal noise from the incoming modes need not be considered.

Carrying out the contractions we find the average $\langle Q \rangle$ and the variance $\mathrm{Var}\, Q = \langle Q^2 \rangle - \langle Q \rangle^2$ of the transferred charge,

$$\langle Q \rangle = \mathrm{Tr}\, \mathcal{P}_- S^\dagger \sigma_z \mathcal{D} \mathcal{P}_+ S, \tag{2.6}$$

$$\mathrm{Var}\, Q = \mathrm{Tr}\, \mathcal{P}_- S^\dagger \mathcal{D} \mathcal{P}_+ S - \mathrm{Tr}\, \mathcal{P}_- S^\dagger \sigma_z \mathcal{D} \mathcal{P}_+ S \mathcal{P}_- S^\dagger \sigma_z \mathcal{D} \mathcal{P}_+ S$$
$$- \mathrm{Tr}\, \mathcal{P}_- S^\dagger \sigma_z \mathcal{D} \mathcal{P}_+ S \mathcal{P}_- S^\dagger \sigma_z \mathcal{D} \mathcal{P}_- S. \tag{2.7}$$

The third term in Eq. (2.7) originates from the anomalous contraction in combination with the particle-hole symmetry relation (2.2). The third term combines with the second term to remove one energy projector,

$$\mathrm{Var}\, Q = \mathrm{Tr}\, \mathcal{P}_- S^\dagger \mathcal{D} \mathcal{P}_+ S - \mathrm{Tr}\, \mathcal{P}_- S^\dagger \sigma_z \mathcal{D} \mathcal{P}_+ S \mathcal{P}_- S^\dagger \sigma_z \mathcal{D} S. \tag{2.8}$$

While Eq. 2.6 for the average charge has an intuitive interpretation of scattering from filled states at $E < 0$ to empty states at $E > 0$, the formula (2.8) for the charge noise is less intuitive. As a check, we show in App. A that it agrees with the more general Klich formula of full counting statistics [25].

# 3 Correspondence between charge variance and average particle number

We apply the general scattering theory to the setting of Fig. 1b. There are $M$ electron-hole modes in each metal contact, $N = 2M$ in total, coupled via a pair of counterpropagating

Majorana edge modes. The coupling is inelastic because of a time dependent phase difference $\phi(t)$ across the Josephson junction that separates the two contacts. The $2\pi$ increment of $\phi$ imposed by a flux bias injects an edge vortex into each contact, and we wish to determine the charge noise associated with that injection process.

The scattering matrix decomposes into transmission blocks $t, t'$ and reflection blocks $r, r'$, each of dimension $M \times M$,

$$S(E, E') = \begin{pmatrix} r(E, E') & t(E, E') \\ t'(E, E') & r'(E, E') \end{pmatrix}. \tag{3.1}$$

The projector

$$\mathcal{D} = \begin{pmatrix} 1 & 0 \\ 0 & 0 \end{pmatrix} \tag{3.2}$$

selects the matrices $t$ and $r$ in the expressions (2.6) and (2.8) for the mean and variance of the charge transferred into the right contact,

$$\langle Q \rangle = \operatorname{Tr} \mathcal{P}_- \big( t^\dagger \sigma_z \mathcal{P}_+ t + r^\dagger \sigma_z \mathcal{P}_+ r \big), \tag{3.3}$$

$$\operatorname{Var} Q = \operatorname{Tr} \mathcal{P}_- \big( t^\dagger \mathcal{P}_+ t + r^\dagger \mathcal{P}_+ r \big) - 2 \operatorname{Re} \operatorname{Tr} \mathcal{P}_- r^\dagger \sigma_z \mathcal{P}_+ t \mathcal{P}_- t^\dagger \sigma_z r$$
$$- \operatorname{Tr} \mathcal{P}_- \big( r^\dagger \sigma_z \mathcal{P}_+ r \mathcal{P}_- r^\dagger \sigma_z r + t^\dagger \mathcal{P}_+ \sigma_z t \mathcal{P}_- t^\dagger \sigma_z t \big). \tag{3.4}$$

We consider the structure of the matrices $t$ and $r$ in more detail.

The $M \times M$ transmission matrix $t(E, E')$ describes propagation from the left contact into the right contact via the right-moving Majorana mode. It can be decomposed as

$$t_{nm}(E, E') = u_n(E) v_m(E') \tau(E, E'), \tag{3.5}$$

in terms of the inelastic transmission amplitude $\tau(E, E')$ of the Majorana mode. The $n = 1, 2, \ldots M$ spinors $u_n(E)$ and $v_n(E)$, normalized to unity,

$$\sum_{n=1}^{M} |u_n(E)|^2 = 1 = \sum_{n=1}^{M} |v_n(E)|^2, \tag{3.6}$$

describe the elastic coupling between the Majorana mode and the electron-hole modes at the interface with the right contact ($u_n$) and the left contact ($v_n$).

The $M \times M$ reflection matrix $r(E, E')$ for reflection of an electron-hole mode incident from the right contact can be decomposed as

$$r_{nm}(E, E') = d_{nm}(E) \delta(E - E') + u_n(E) w_m(E') \rho(E, E'). \tag{3.7}$$

The first term $d_{nm}$ describes direct elastic reflection at the interface between the superconductor and the right contact. The second term describes inelastic reflection at the Josephson junction, decomposed as the product of the transmission amplitude $w_m$ from the right contact into the left-moving Majorana mode, the reflection amplitude $\rho$ from the Josephson junction, and the transmission amplitude $u_n$ from the right-moving Majorana mode into the right contact. Both $u_n$ and $w_m$ are normalized to unity. Note that $u_n$ appears also in the decomposition (3.5) of $t_{nm}$.

We make the key assumption that the elastic scattering at the superconductor–contact interface is only weakly energy dependent near the Fermi level, $E = 0$, so that we may approximate $u_n(E) \approx u_n(0)$.

To justify this approximation, we note, on the one hand, that the characteristic energy dependence of the elastic scattering amplitudes is on the scale of $E_{\text{elastic}} \simeq \hbar v_{\text{F}}/\xi_0$, where $v_{\text{F}}$ is the Fermi velocity and the superconducting coherence length $\xi_0$ sets the effective width of the interface. On the other hand, the characteristic energy dependence of the inelastic scattering by the Josephson junction is on the scale $E_{\text{inelastic}} = \hbar(W/\xi_0)\dot{\phi}$, where $W$ is the junction width and $\dot{\phi}$ the rate of change of the superconducting phase [17]. It is consistent to neglect the energy dependence of $u_n(E)$ while retaining the energy dependence of $\tau(E, E')$ and $\rho(E, E')$ if $E_{\text{inelastic}} \ll E_{\text{elastic}}$, hence if the junction is sufficiently narrow:

$$E_{\text{inelastic}} \ll E_{\text{elastic}} \Rightarrow W \ll v_{\text{F}}/\dot{\phi}. \tag{3.8}$$

As we show in App. B, this single assumption combined with particle-hole symmetry implies that the following matrix products vanish:

$$\begin{aligned}
\mathcal{P}_- t^\dagger \sigma_z \mathcal{P}_+ t \mathcal{P}_- &= 0, \\
\mathcal{P}_- r^\dagger \sigma_z \mathcal{P}_+ r \mathcal{P}_- &= 0, \\
\mathcal{P}_- r^\dagger \sigma_z \mathcal{P}_+ t \mathcal{P}_- &= 0.
\end{aligned} \tag{3.9}$$

What underlies these three identities is that the inelastic contributions to the transmission and reflection matrices are rank-one matrices in the mode index.

It follows upon combination of Eqs. (3.3) and (3.9), and noting that $\text{Tr}\,\mathcal{P}_-(\cdots) = \text{Tr}\,\mathcal{P}_-(\cdots)\mathcal{P}_-$, that there is no charge transfer into the right contact on average,

$$\langle Q \rangle = 0. \tag{3.10}$$

For the charge noise (3.4), Eq. (3.9) implies that the second and third trace vanish, only the first trace remains:

$$\begin{aligned}
\text{Var}\,Q &= e^2 \,\text{Tr}\,\mathcal{P}_- \big( t^\dagger \mathcal{P}_+ t + r^\dagger \mathcal{P}_+ r \big) \mathcal{P}_- \\
&= e^2 \int_0^\infty \frac{dE}{2\pi} \int_{-\infty}^0 \frac{dE'}{2\pi} \big( |\tau(E, E')|^2 + |\rho(E, E')|^2 \big).
\end{aligned} \tag{3.11}$$

Eq. (3.11) states that the charge variance (divided by $e^2$) equals the average number of quasiparticles injected into the right contact by the time dependent phase difference across the Josephson junction. This relationship is analogous to the known relationship between electrical shot noise and thermal conductance in a setting without time-dependent driving [21, 23, 24].

# 4 Evaluation of the charge noise

We evaluate Eq. (3.11) for the case that the phase difference $\phi$ across the junction is advanced at a constant rate $\dot{\phi} = 2\pi/T$, via a linearly increasing flux bias $\Phi(t) = (h/2e)t/T$. We work in the adiabatic regime that the propagation time $\tau_W = W/v_{\text{F}}$ along the Josephson junction is small compared to the inelastic scattering time,

$$\tau_W \ll \hbar/E_{\text{inelastic}} \Rightarrow W \ll (\xi_0/W)v_{\text{F}}/\dot{\phi}. \tag{4.1}$$

The adiabaticity condition is stronger than the earlier assumption (3.8) for $W > \xi_0$.

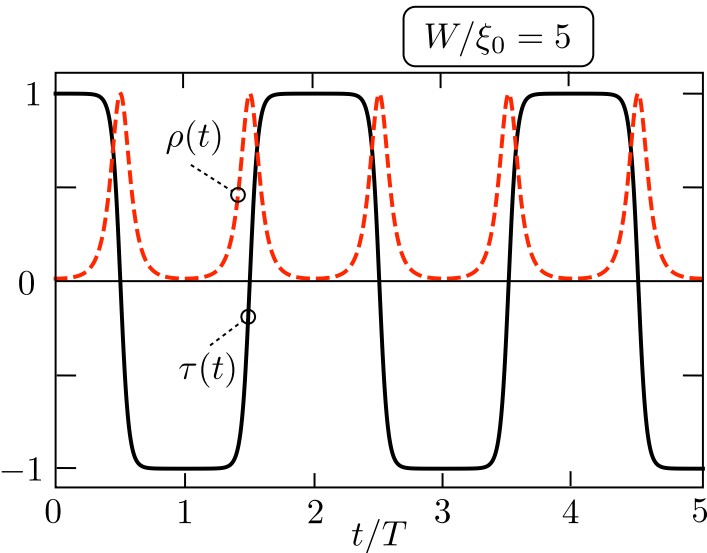

Figure 2: Plot of the transmission and reflection amplitudes (4.4), calculated for a linearly increasing phase difference $\phi(t) = 2\pi t/T$ across the Josephson junction. The junction fully reflects the counterpropagating Majorana edge modes when $\phi = \pi$ modulo $2\pi$.

The adiabatic scattering matrix depends only on the energy difference,

$$S(E, E') = \int_{-\infty}^{\infty} dt \, e^{i(E-E')t} S(t), \tag{4.2}$$

it is the Fourier transform of the "frozen" scattering matrix $S(t)$ — evaluated for fixed value $\phi(t)$ of the superconducting phase difference. The transmission and reflection amplitudes $\tau(E, E') = \tau(E - E')$ and $\rho(E, E') = \rho(E - E')$ are likewise the Fourier transform of the "frozen" counterparts $\tau(t)$ and $\rho(t)$.

The adiabatic scattering matrix of a Josephson junction between counterpropagating edge modes is given by [8]

$$S(t) = \begin{pmatrix} 1/\cosh\beta(t) & \tanh\beta(t) \\ \tanh\beta(t) & -1/\cosh\beta(t) \end{pmatrix}, \quad \beta(t) = \frac{W}{\xi_0}\cos(\pi t/T). \tag{4.3}$$

The corresponding transmission and reflection amplitudes

$$\tau(t) = \tanh\beta(t), \quad \rho(t) = 1/\cosh\beta(t) \tag{4.4}$$

are plotted in Fig. 2. The transmission amplitude is periodic with period $2T$, twice the period of the superconducting phase $\phi(t)$ because a $2\pi$ increment of $\phi$ is a $\pi$ increment of the fermionic phase.

We write the charge noise formula (3.11) in the time domain, with a detection window $(0, 2\mathcal{N}T)$ that is a multiple of the periodicity $2T$,

$$\operatorname{Var} Q = -\frac{e^2}{4\pi^2} \int_0^{2\mathcal{N}T} dt \int_0^{2\mathcal{N}T} dt' \, \frac{\tau(t)\tau(t') + \rho(t)\rho(t')}{(t - t' + i\epsilon)^2}. \tag{4.5}$$

The singularity at $t = t'$ is regularized by the infinitesimal $\epsilon > 0$. The charge noise per vortex is

$$\text{Var}\, Q_{\text{vortex}} = \frac{1}{2} \lim_{\mathcal{N} \to \infty} \frac{1}{\mathcal{N}} \text{Var}\, Q, \tag{4.6}$$

the factor of $1/2$ is there because two vortices are injected into each edge in a time $2T$.

In view of the periodicity $\tau(t + 2T) = \tau(t)$, $\rho(t + 2T) = \rho(t)$ we have

$$
\begin{aligned}
\text{Var}\, Q_{\text{vortex}} &= -\lim_{\mathcal{N} \to \infty} \frac{e^2}{8\mathcal{N}\pi^2} \sum_{n=0}^{\mathcal{N}} \sum_{m=0}^{\mathcal{N}} \int_0^{2T} dt \int_0^{2T} dt' \, \frac{\tau(t)\tau(t') + \rho(t)\rho(t')}{(t - t' + 2T(n - m) + i\epsilon)^2} \\
&= -\frac{e^2}{32T^2} \int_0^{2T} dt \int_0^{2T} dt' \, \frac{\tau(t)\tau(t') + \rho(t)\rho(t')}{\sin^2\left[\frac{1}{2}(\pi/T)(t - t' + i\epsilon)\right]} \\
&= -\frac{e^2}{32\pi^2} \int_0^{2\pi} dt \int_0^{2\pi} dt' \, \frac{\sinh\left(\frac{W}{\xi_0} \cos t\right) \sinh\left(\frac{W}{\xi_0} \cos t'\right) + 1}{\sin^2\left[\frac{1}{2}(t - t' + i\epsilon)\right] \cosh\left(\frac{W}{\xi_0} \cos t\right) \cosh\left(\frac{W}{\xi_0} \cos t'\right)}.
\end{aligned}
\tag{4.7}
$$

Because of the identity

$$\int_0^{2\pi} dt \int_0^{2\pi} dt' \, \frac{1}{\sin^2\left[\frac{1}{2}(t - t' + i\epsilon)\right]} = 0, \tag{4.8}$$

we may rewrite the integral (4.7) as

$$\text{Var}\, Q_{\text{vortex}} = -\frac{e^2}{32\pi^2} \int_0^{2\pi} dt \int_0^{2\pi} dt' \, \frac{1 - \cosh\left(\frac{W}{\xi_0}(\cos t - \cos t')\right)}{\sin^2\left[\frac{1}{2}(t - t')\right] \cosh\left(\frac{W}{\xi_0} \cos t\right) \cosh\left(\frac{W}{\xi_0} \cos t'\right)}. \tag{4.9}$$

The infinitesimal $\epsilon$ may now be set to zero, the integral remains finite.

The $W$-dependence of $\text{Var}\, Q_{\text{vortex}}$ is plotted in Fig. 3. The asymptotics for small and for large $W/\xi_0$ are[1]

$$
\begin{aligned}
\text{Var}\, Q_{\text{vortex}} &= \frac{e^2}{8} (W/\xi_0)^2 \ \text{ for } \ W/\xi_0 \ll 1, \\
\text{Var}\, Q_{\text{vortex}} &= \frac{e^2}{\pi^2} \ln(2\pi W/\xi_0) \ \text{ for } \ W/\xi_0 \gg 1.
\end{aligned}
\tag{4.10}
$$

The large-$W$ asymptotics can be written equivalently as Eq. (1.1), with a logarithmic dependence on the ratio of the separation $L = 2\pi v_{\text{F}}/\dot\phi$ between subsequent edge vortices and the width $\lambda = (v_{\text{F}}/\dot\phi)(\xi_0/W)$ of the phase boundary which represents the core of the edge vortex.[2]

## 5 Discussion

The experimental observable in a shot noise measurement is the noise power $P$, being the correlator of the time dependent current fluctuations $\delta I(t)$:

$$P = \int_{-\infty}^{\infty} dt \, \langle \delta I(0) \delta I(t) \rangle = \lim_{t \to \infty} \frac{1}{t} \left( \langle Q(t)^2 \rangle - \langle Q(t) \rangle^2 \right). \tag{5.1}$$

---

[1]For the small-$W$ asymptotics, expansion of the integrand in Eq. (4.9) to second order in $W/\xi_0$ gives $(W/\xi_0)^2[\cos(t + t') - 1]$, which is then readily integrated. For the large-$W$ asymptotics, see App. C.

[2]The time $\lambda/v_{\text{F}} = \hbar/E_{\text{inelastic}}$ is the width of the peaks in $\rho(t)$ in Fig. 2.

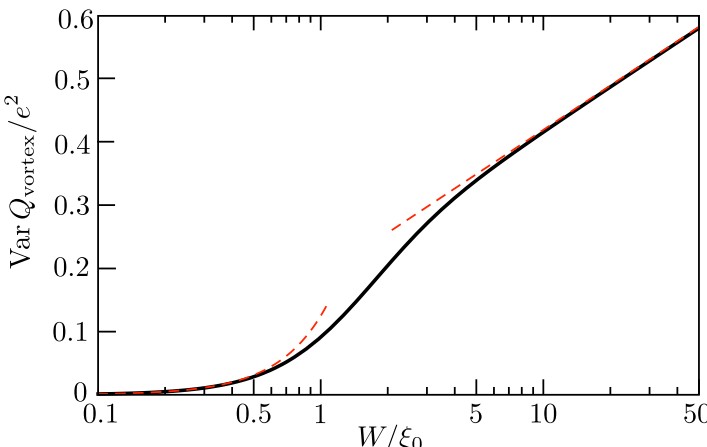

Figure 3: Plot of the charge noise per vortex as a function of the ratio $W/\xi_0$ (logarithmic scale). The solid curve is computed from Eq. (4.9), the dashed curves are the asymptotes (4.10).

Here $Q(t)$ is the transferred charge in a time $t$.

For the flux-biased vortex injector of Fig. 1b the result (1.1) implies that

$$P_{\text{vortex}} = \frac{1}{T} \operatorname{Var} Q_{\text{vortex}} = \frac{e^2}{h} \frac{2e\dot{\Phi}}{\pi^2} \ln(L/\lambda), \quad \text{for} \quad L \gg \lambda. \tag{5.2}$$

We contrast this with the shot noise power of the fermion injector of Fig. 1a, given by [23]

$$P_{\text{fermion}} = \frac{e^2}{h} \frac{eV}{2}. \tag{5.3}$$

A flux rate of change $\dot{\Phi}$ is equivalent to a voltage bias $V$, so the replacement $\dot{\Phi} \leftrightarrow V$ in the two formulas is expected. The key difference is the appearance of a logarithmic dependence of the vortex shot noise on the separation of subsequent vortices. There is no such dependence on the Majorana fermion separation. This nonlocality suggests that an unpaired edge vortex has a divergent charge noise, which indeed it does (see App. D).

To observe the anomalous dependence of $P_{\text{vortex}}$ on the edge vortex separation, one would need to be able to vary the ratio $L/\lambda$. In the geometry of Fig. 1b one has $L/\lambda = 2\pi W/\xi_0$, so this ratio is fixed by the parameters of the Josephson junction. Since it might be problematic to engineer a junction with adjustable width, we show in Fig. 4 an alternative double-junction geometry where the ratio $L/\lambda$ can be varied at a fixed geometry by a voltage bias.

A $2\pi$ increment of $\phi$ injects two vortices on each edge, one for each Josephson junction. The separation $L$ of the edge vortices now equals the spacing between the two Josephson junctions, so this length is fixed by the geometry. However, the vortex core size $\lambda = (v_{\text{F}}/\dot{\phi})(\xi_0/W) = (hv_{\text{F}}/2eV)(\xi_0/W)$ can be adjusted by varying the voltage bias $V$, allowing for a measurement of the anomalous $L/\lambda$ dependence of the shot noise power in a fixed geometry. The resulting logarithmic voltage dependence of the shot noise power,[3]

$$P_{\text{vortex}} = \frac{e^2}{h} \frac{4eV}{\pi^2} \ln(V/V_c), \quad V_c = \frac{\hbar v_{\text{F}} \xi_0}{2eLW}, \tag{5.4}$$

---

[3]The calculation of the charge variance for the geometry of Fig. 4 is worked out in App. E.

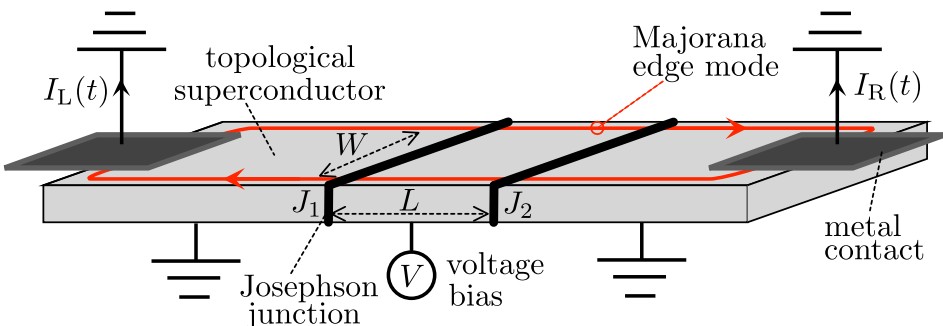

Figure 4: Variation on the geometry of Fig. 1b, with two Josephson junctions instead of a single junction, and a voltage bias instead of a flux bias. The shot noise power increases as $V \ln V$ with the applied voltage.

holds over wide voltage range $V_c \ll V \ll (W/\xi_0)V_c$ for $W \gg \xi_0$. This $V \ln V$ increase of $P_{\text{vortex}}$ contrasts with the purely linear voltage dependence of $P_{\text{fermion}}$ and serves as a distinguishing signature between these two types of excitations of a Majorana edge mode, a signature that is accessible by a purely electrical transport measurement.

## Acknowledgements

This project has received funding from the Netherlands Organization for Scientific Research (NWO/OCW) and from the European Research Council (ERC) under the European Union's Horizon 2020 research and innovation programme. We have benefited from discussions with I. Adagideli.

## A    Consistency of Eq. (2.8) with the Klich formula for the cumulant generating function

In the main text we derived the formula (2.8) for the variance of the transmitted charge directly from the contractions (2.5). We showed that the anomalous contraction of two creation operators has the effect of eliminating one of the projectors onto positive energies. As a check, we show here how the same result follows from the Klich formula [25] in the theory of full counting statistics.

We note the sequence of equalities

$$\text{Var} \, Q = \text{Tr} \, \mathcal{P}_- S^\dagger \mathcal{D} \mathcal{P}_+ S - \text{Tr} \, \mathcal{P}_- S^\dagger \sigma_z \mathcal{D} \mathcal{P}_+ S \mathcal{P}_- S^\dagger \sigma_z \mathcal{D} S$$
$$= \text{Tr} \, \mathcal{P}_- S^\dagger \sigma_z \mathcal{D} \mathcal{P}_+ S \mathcal{P}_+ S^\dagger \sigma_z \mathcal{D} S$$
$$= \text{Tr} \, \mathcal{P}_- S^\dagger \sigma_z \mathcal{D} \mathcal{P}_- S \mathcal{P}_+ S^\dagger \sigma_z \mathcal{D} S. \tag{A.1}$$

For the second equality we substituted $S\mathcal{P}_- S^\dagger = 1 - S\mathcal{P}_+ S^\dagger$ and used $(\sigma_z \mathcal{D})^2 = \mathcal{D}$. The third equality follows from particle-hole symmetry.[4] Hence, by adding the second and third

---

[4]The particle-hole symmetry relation (2.2) of the scattering matrix implies that traces of the form (A.1) are invariant upon the replacements: $\text{Tr} \, M \mapsto \text{Tr} \, M^\dagger$, $\sigma_z \mapsto -\sigma_z$, $\mathcal{P}_\pm \mapsto \mathcal{P}_\mp$.

equality we arrive at

$$\text{Var}\, Q = \tfrac{1}{2}\,\text{Tr}\,\mathcal{P}_- S^\dagger \sigma_z \mathcal{D} S \mathcal{P}_+ S^\dagger \sigma_z \mathcal{D} S. \tag{A.2}$$

Each factor $\sigma_z \mathcal{D}$ now appears without an energy projector. Similarly, the expression (2.6) for the average charge can be rewritten identically as[5]

$$\langle Q \rangle = \tfrac{1}{2}\,\text{Tr}\,\mathcal{P}_- S^\dagger \sigma_z \mathcal{D} S, \tag{A.3}$$

without the energy projector multiplying $\sigma_z \mathcal{D}$.

Eqs. (A.2) and (A.3) agree with the Klich formula for the cumulant generating function[6] [20]

$$\ln\langle e^{i\xi Q} \rangle = \tfrac{1}{2}\ln \text{Det}\left[1 - \mathcal{P}_- + \mathcal{P}_- S^\dagger e^{i\xi \sigma_z \mathcal{D}} S\right]$$
$$= \tfrac{1}{2}i\xi\,\text{Tr}\,\mathcal{P}_- S^\dagger \sigma_z \mathcal{D} S - \tfrac{1}{4}\xi^2\,\text{Tr}\,\mathcal{P}_- S^\dagger \sigma_z \mathcal{D} S \mathcal{P}_+ S^\dagger \sigma_z \mathcal{D} S + \mathcal{O}(\xi^3). \tag{A.4}$$

# B  Proof of Eq. (3.9)

To show that the three matrix products (3.9) all vanish, we substitute the decompositions (3.5) and (3.7) of the transmission and reflection matrices. Because the reflection matrix in Eq. (3.9) is sandwiched between projectors $\mathcal{P}_+$ and $\mathcal{P}_-$, the elastic contribution $d_{nm}$ in Eq. (3.7) drops out. The inelastic contributions to each matrix product contain the same factor

$$\sum_{n=1}^{M} u_n^\dagger(E)\sigma_z u_n(E) = \sum_{n=1}^{M} u_n^\text{T}(-E)(\sigma_x \cdot \sigma_z)u_n(E) = -i\sum_{n=1}^{M} u_n^\text{T}(-E)\sigma_y u_n(E), \tag{B.1}$$

where in the second equality we used particle-hole symmetry.

We now make the assumption, valid for $W \ll v_\text{F}/\dot{\phi}$, that we can neglect the energy dependence of the elastic coupling amplitude $u_n(E) \approx u_n(0)$ between the right-moving Majorana mode and the right contact. Then Eq. (B.1) reduces to zero because $\sigma_y$ is an antisymmetric matrix, hence $u_n^T \sigma_y u_n = 0$.

---

[5]Eq. (A.3) follows from Eq. (2.6) in view of equalities $\text{Tr}\,\mathcal{P}_- S^\dagger \sigma_z \mathcal{D} \mathcal{P}_+ S = -\text{Tr}\,\mathcal{P}_+ S^\dagger \sigma_z \mathcal{D} \mathcal{P}_+ S = \text{Tr}\,\mathcal{P}_+ S^\dagger \sigma_z \mathcal{D} \mathcal{P}_- S$. The first equality holds because $\text{Tr}\,S^\dagger \sigma_z \mathcal{D} \mathcal{P}_+ S = 0$, the second equality follows from particle-hole symmetry.

[6]In Eq. (3.12) of Ref. 20 the generating function contains a $\sigma_y$ instead of a $\sigma_z$ Pauli matrix, because there the Majorana basis instead of the electron-hole basis is chosen for the Nambu spinors.

## C   Computation of the logarithmic asymptote of the charge noise

To derive the logarithmic large-$W$ asymptotics of Eq. 4.10, we note that for $W \gg \xi_0$ the scattering amplitude profile (4.3) is well described by the approximation [17]

$$\tau(t) = \begin{cases} -\tanh[\frac{1}{2}(t - T/2)/t_0] & \text{for } 0 < t < T, \\ \tanh[\frac{1}{2}(t - 3T/2)/t_0] & \text{for } T < t < 2T, \end{cases} \tag{C.1a}$$

$$\rho(t) = \begin{cases} 1/\cosh[\frac{1}{2}(t - T/2)/t_0] & \text{for } 0 < t < T, \\ 1/\cosh[\frac{1}{2}(t - 3T/2)/t_0] & \text{for } T < t < 2T, \end{cases} \tag{C.1b}$$

$$t_0 = (\xi_0/W)(T/2\pi), \tag{C.1c}$$

repeated periodically with period $2T$. On the scale of Fig. 2, with $W/\xi_0 = 5$, the approximation is nearly indistinguishable from the full result.

The Fourier coefficients

$$\tau(\omega_n) = \int_0^{2T} dt\, e^{i\omega_n t} \tau(t), \quad \rho(\omega_n) = \int_0^{2T} dt\, e^{i\omega_n t} \rho(t), \quad \omega_n = \pi n/T, \tag{C.2}$$

in the large-$W/\xi_0$ regime can be calculated from the integrals

$$\int_{-\infty}^{\infty} dt\, e^{i\omega t} \tanh(\tfrac{1}{2}t/t_0) = \frac{2\pi i t_0}{\sinh(\pi\omega t_0)}, \\ \int_{-\infty}^{\infty} dt\, e^{i\omega t} \frac{1}{\cosh(\tfrac{1}{2}t/t_0)} = \frac{2\pi t_0}{\cosh(\pi\omega t_0)}, \tag{C.3}$$

with the result

$$\tau(\omega_n) = \left(e^{i\omega_n T/2} - e^{i\omega_n 3T/2}\right) \frac{2\pi i t_0}{\sinh(\pi\omega_n t_0)} \Rightarrow |\tau(\omega_n)|^2 = \delta_{n,\text{odd}} \frac{(4\pi t_0)^2}{\sinh^2(\pi\omega_n t_0)}, \\ \rho(\omega_n) = \left(e^{i\omega_n T/2} + e^{i\omega_n 3T/2}\right) \frac{2\pi t_0}{\cosh(\pi\omega_n t_0)} \Rightarrow |\rho(\omega_n)|^2 = \delta_{n,\text{even}} \frac{(4\pi t_0)^2}{\cosh^2(\pi\omega_n t_0)}. \tag{C.4}$$

The charge noise per vortex then follows by writing Eq. (3.11) as a Fourier series,

$$\text{Var}\, Q_{\text{vortex}} = \frac{e^2}{4\pi^2} \frac{\pi}{2T} \sum_{n=0}^{\infty} \omega_n \left(|\tau(\omega_n)|^2 + |\rho(\omega_n)|^2\right). \tag{C.5}$$

For $T/t_0 = 2\pi W/\xi_0 \gg 1$ the sum may be approximated by an integral and produces the logarithmic growth

$$\text{Var}\, Q_{\text{vortex}} \to \frac{e^2}{\pi^2} \ln(T/t_0), \quad \text{for } T \gg t_0. \tag{C.6}$$

# D   Divergent charge noise for an unpaired edge vortex

If a single vortex is injected into each edge, the scattering amplitudes (C.1) in the time interval $(0, T)$ hold for all times,

$$
\tau(t) = -\tanh(\tfrac{1}{2}t/t_0) \Rightarrow \tau(E, E') = -\frac{2\pi i t_0}{\sinh[\pi(E - E')t_0]},
$$

$$
\rho(t) = 1/\cosh(\tfrac{1}{2}t/t_0) \Rightarrow \rho(E, E') = \frac{2\pi t_0}{\cosh[\pi(E - E')t_0]}.
$$

(D.1)

Substitution into Eq. (3.11) gives an expression for the charge noise,

$$
\operatorname{Var} Q = e^2 t_0^2 \int_0^\infty dE\, E \left( \frac{1}{\sinh^2 \pi E t_0} + \frac{1}{\cosh^2 \pi E t_0} \right),
$$

(D.2)

with a logarithmic divergence at $E = 0$.

For a finite answer we may introduce a finite detection time $t_{\text{det}}$, cutting off the integral for $E \lesssim 1/t_{\text{det}}$, which gives

$$
\operatorname{Var} Q = \frac{e^2}{\pi^2} \ln(t_{\text{det}}/t_0), \quad \text{for } t_{\text{det}} \gg t_0.
$$

(D.3)

In the case of a periodic sequence of edge vortices considered in the main text, the spacing $T$ between subsequent vortices takes over from $t_{\text{det}}$ to provide a finite charge variance.

# E   Charge noise in a double-Josephson junction geometry

In Fig. 4 we have modified the geometry of Fig. 1b to include a second Josephson junction next to the first. A flux bias, or equivalently a voltage bias as in the figure, will then inject two edge vortices on each edge.

The scattering matrix of the pair of Josephson junctions is composed from the scattering matrices $S_{J_1}$, $S_{J_2}$ of the individual junctions, for which we take the adiabatic approximation,

$$
S_{J_n}(E, E') = \int_{-\infty}^\infty dt\, e^{i(E - E')t} S_{J_n}(t),
$$

$$
S_{J_n}(t) = \begin{pmatrix} \sin\alpha_n(t) & \cos\alpha_n(t) \\ \cos\alpha_n(t) & -\sin\alpha_n(t) \end{pmatrix}, \quad \alpha_n(t) = \arccos\tanh\beta(t).
$$

(E.1)

Adiabaticity requires that the time $W/v_{\text{F}}$ to move from one edge to the opposite edge along a junction is short compared to the vortex injection time $t_0 = (\xi_0/W)\dot\phi^{-1}$. The time $L/v_{\text{F}}$ to move from one junction to the next may be large compared to $t_0$.

The phase fields $\alpha_1(t)$ and $\alpha_2(t)$ of the two Josephson junctions both switch from 0 to $\pi$ on a time scale $t_0$ around $t = 0$.[7] If $\lambda = v_{\text{F}}t_0 \ll L$ the two edge vortices injected by these

---

[7] For counterpropagating edge modes the phase $\alpha$ is an *even* function of the phase difference $\phi$ across the Josephson junction [8]. For co-propagating edge modes, in contrast, $\alpha$ is an *odd* function of $\phi$ and in that case $\alpha_1$ and $\alpha_2$ would have opposite sign [17].

switching events do not overlap. We consider that regime in what follows and for ease of notation set $v_{\mathrm{F}} \equiv 1$.

The transmission amplitude $\tau(E, E')$ from left to right and the reflection amplitude $\rho(E, E')$ from the right are given in the time domain by

$$
\begin{aligned}
\tau(t, t') &= \delta(t - t' - L) \cos \alpha_2(t' + L) \cos \alpha_1(t'), \\
\rho(t, t') &= \delta(t - t') \sin \alpha_2(t) + \delta(t - t' - 2L) \cos \alpha_2(t' + 2L) \sin \alpha_1(t' + L) \cos \alpha_2(t').
\end{aligned}
\tag{E.2}
$$

The assumption $L \gg \lambda$ prevents the appearance of terms delayed by more than $2L$, or equivalently, there are no multiple reflections at the junctions.

Using again that $L \gg \lambda$ we note that $\cos \alpha_2(t' + 2L) \cos \alpha_2(t') \approx -1$ whenever $\sin \alpha_1(t' + L)$ is nonzero, hence we may simplify the expression for $\rho$ into

$$
\rho(t, t') = \delta(t - t') \sin \alpha_2(t) - \delta(t - t' - 2L) \sin \alpha_1(t' + L).
\tag{E.3}
$$

At the same level of approximation, we have

$$
\tau(t, t') = \delta(t - t' - L)[\cos \alpha_2(t' + L) - \cos \alpha_1(t') + 1].
\tag{E.4}
$$

Transformation to the energy domain gives

$$
\begin{aligned}
\tau(E, E') &= e^{iE'L} \left[ c_2(E - E') - e^{i(E - E')L} c_1(E - E') + 2\pi \delta(E - E') \right], \\
\rho(E, E') &= s_2(E - E') - e^{i(E + E')L} s_1(E - E'),
\end{aligned}
\tag{E.5}
$$

with the definitions

$$
c_n(E) = \int_{-\infty}^{\infty} dt \, e^{iEt} \cos \alpha_n(t), \quad s_n(E) = \int_{-\infty}^{\infty} dt \, e^{iEt} \sin \alpha_n(t).
\tag{E.6}
$$

The dominant contribution to the charge noise in Eq. (3.11) comes from the transmission amplitude, because of the $1/E$ singularity of $c_1(E)$ and $c_2(E)$ according to Eq. (D.1). For the single-vortex noise we needed a finite detection time to cut off the singularity, here the spacing $L$ of the vortices is an effective cut-off in the case $c_1 = c_2$ of two identical tunnel junctions. Then we find

$$
\operatorname{Var} Q \approx e^2 \lambda^2 \int_0^{\infty} dE \, E \, \frac{|1 - e^{iEL}|^2}{\sinh^2 \pi E \lambda} \to \frac{2e^2}{\pi^2} \ln(L/\lambda), \quad \text{for } L \gg \lambda.
\tag{E.7}
$$

This is twice the result (1.1) because it refers to two vortices.

A constant applied voltage $V$ cause the superconducting phase to increase linearly in time, $\dot{\phi} = 2eV/\hbar$, hence $\lambda = v_{\mathrm{F}}(\xi_0/W)(\hbar/2eV)$. If $V \ll \hbar v_{\mathrm{F}}/eL$ the injected edge vortices from subsequent periods do not overlap. The resulting shot noise power $P = (\dot{\phi}/2\pi) \operatorname{Var} Q$ takes the form

$$
P = \frac{e^2}{h} \frac{4eV}{\pi^2} \ln \left( \frac{2eV L W}{\hbar v_{\mathrm{F}} \xi_0} \right), \quad \text{for } \frac{\hbar v_{\mathrm{F}}}{L} \frac{\xi_0}{W} \ll eV \ll \frac{\hbar v_{\mathrm{F}}}{L}.
\tag{E.8}
$$

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
