# Peer review of "Shot noise distinguishes Majorana fermions from vortices injected in the edge mode of a chiral p-wave superconductor"

_SciPost Physics_

## Round 1 · Referee Report · Anonymous (Referee 1) · 2020-10-7

Strengths

1-timely subject matter
2-clear, concise presentation
3-clear statement with proposal for experimental verification

Weaknesses

1-connection between language used in introduction and the calculation itself cann be improved
2-an essential extension of the result is discussed in the appendix

Report

Beenakker and Oriekhov consider the charge noise of a flux-biased Josephson junction between two chiral p-wave superconductors for the case that the flux bias increases linearly in time. They find that in a suitably defined adiabatic limit the flux noise depends logarithmically on the junction width W. For comparison, the charge noise for a superconductor edge weakly coupled to a voltage-biased normal metal does not have such a logarithmic factor. This difference is interpreted as the difference between the charge noise of Majorana edge modes and edge vortices.

The calculations employ the formalism of time-dependent scattering theory. The details of the calculation are explained in a concise, but clear manner. The approximations required to arrive at the final result are discussed in detail. The subject matter is timely. The article has a clear message that will be of interest to specialists in the field. I recommend publication in SciPost Physics.

Requested changes

While the presentation is already very good as it is, I have a number of suggestions that may help to make it even better. Also, I have a few questions regarding appendices D and E, the results of which are mentioned in Sec. 4 of the main text.

  1. In the introduction "edge" vortices are introduced and it is mentioned that they are "mobile". The calculations are for a Josephson junction. Although a connection between the JJ and an edge vortex can be drawn, whatever edge vortices exist in the calculation are clearly localized at the junction and not able to move along the edge. It would be helpful if the connection between the language used in the introduction and the calculation could be strengthened.

  2. The article repeatedly mentions "inelastic scattering". This may be confusing, because "inelastic" may also be used to refer to quasiparticle collisions and other many-body effects. In the present case, the formulation is for scattering from a time-dependent potential.

  3. For the problem at hand, for which the phase factor exp(i phi) across the Josephson junction is a periodic function of the time with period T, the energy difference E-E' is an integer multiple of (half) the frequency omega = 2 pi/T. It may be helpful to point this out in when the energy-dependent scattering matrix S(E,E') is introduced in Sec. 2 or in Sec. 3.

  4. Directly below Eq. (2.2): The authors seek the charge transferred by ... if the incoming modes are in thermal equilibrium.

  5. The projector "D_M" is called "D" in most of the article. Perhaps it is better to call it "D" everywhere.

  6. The words "energy dependence" appear twice on the top of page 6, but their meaning is subtly different. For the amplitudes u(E) "energy dependence" refers to the energy scale on which they vary with E. For the scattering matrix S(E,E') it is the typical energy difference |E-E'| that is accumulated upon scattering. The equivalent of the "energy dependence" as it is used for u(E) would be a dependence on the center energy (E+E')/2, which is neglected altogether.

  7. For readers familiar with the physics of Josephson junctions, but not with the formalism used by Beenakker and Oriekhov, it may be helpful to point out that the energy quanta hbar omega that are exchanged upon reflection from the JJ are the same as the energy quanta exchanged in multiple Andreev reflections at a flux/voltage biased junction.

  8. Equation (4.3) is the frozen scattering matrix, not the adiabatic scattering matrix.

  9. Section 5 mentions that a single edge vortex has a divergent charge noise. In appendix D a finite observation time is introduced to cut off the divergence. I would expect that the various approximations made in the article (e.g., the adiabatic approximation) could also cut off the divergence. If that is correct, what is the variance of the charge in the limit of an infinite observation time?

  10. The approximations made for the calculations in this article (adiabaticity, neglecting E dependence of u(E)) also pose restrictions on V, L, and W for the double-junction geometry of Fig. 4. What are these? Do they make it difficult to observe the logarithmic dependence on V? I guess the answer is given in the small print of appendix E, but since equation (5.4) makes it into the main text, a discussion of its condition of validity is in the main text would be helpful.

  11. It appears the phase differences across the two junctions in the geometry of Fig. 4 are synchronized, see equation (E.1). This not discussed in the text, which only mentions that there are equal voltage drops over the two junctions. This, however, only requires that the phase differences have the same time derivative, but still allows for a constant shift. How does a constant shift affect the result? Can it drift in time? How can this shift be controlled experimentally?

---

## Round 1 · Referee Report · Anonymous (Referee 2) · 2020-10-27

Report

In this theoretical paper, the authors calculate the shot noise power of a current of edge vortices in a chiral p-wave superconductor in different device geometries. Surprisingly, the noise power increases with the vortices' separation L, even if L >> lamda where lambda is the size of the vortex core. For the device shown in Fig. 4, lambda can be controlled by a voltage bias V (whereas L is fixed by the geometry). In this case, the predicted effect results in a V ln V behavior of the noise power.

The authors stress repeatedly that the vortex noise power differs from the Majorana shot noise power (that is linear in V), but can this difference be probed in *one* device, or is the idea to compare the experimental results obtained in the two devices shown in Fig. 1(a) and Fig. 1(b)/Fig. 4?

The paper is clearly written and predicts an interesting effect in a timely field.
In my opinion, it is suitable for publication in SciPost once the point mentioned above is clarified.

---

## Editorial Decision

resubmitted